# Effects of Intraoperative Magnesium and Ketorolac on Catheter-Related Bladder Discomfort after Transurethral Bladder Tumor Resection: A Prospective Randomized Study

**DOI:** 10.3390/jcm11216359

**Published:** 2022-10-27

**Authors:** Jung-Woo Shim, Seunghee Cha, Hyong Woo Moon, Young Eun Moon

**Affiliations:** 1Department of Anesthesiology and Pain Medicine, Seoul St. Mary’s Hospital, College of Medicine, The Catholic University of Korea, Seoul 06591, Korea; 2Department of Anesthesiology and Pain Medicine, Yeouido St. Mary’s Hospital, College of Medicine, The Catholic University of Korea, Seoul 06591, Korea; 3Department of Urology, Seoul St. Mary’s Hospital, College of Medicine, The Catholic University of Korea, Seoul 06591, Korea

**Keywords:** magnesium, ketorolac, catheter-related bladder discomfort, transurethral resection of bladder tumor

## Abstract

Transurethral resection of bladder tumor (TURBT) is a standard treatment for non-muscle invasive bladder cancer. However, catheter-related bladder discomfort (CRBD) often occurs due to bladder irritation caused by indwelling large-diameter urinary catheters and delays patient recovery. We investigated the efficacy of the intraoperative administration of magnesium and ketorolac in preventing early CRBD after TURBT. One hundred patients scheduled for TURBT were enrolled in this prospective, randomized, double-blind study from December 2021 to June 2022. During surgery, the experimental group (*n* = 48) received intravenous magnesium and ketorolac, and the control group (*n* = 50) received only intravenous ketorolac. The primary outcome was CRBD incidence immediately after surgery. CRBD severity and the postoperative recovery profiles were also evaluated. Compared to the control group, the experimental group had significantly less CRBD until 1 h after surgery (0 h: 31.3% vs. 52.0%, *p* = 0.037; 1 h: 54.2% vs. 74.0%, *p* = 0.041). However, the two groups did not differ in other postoperative findings, including CRBD severity. Co-administration of magnesium and ketorolac during surgery significantly decreased the incidence of early CRBD after TURBT compared to the single use of ketorolac.

## 1. Introduction

Over the past decade, the prevalence of bladder cancer has significantly increased [1], and approximately 75% of newly diagnosed bladder cancer are classified as non-muscle invasive bladder cancer [2]. Although the pathogenesis of bladder cancer has not been fully established yet, metabolomic analysis to reveal the aberrant metabolic pathway is suggested as the potential tumor markers for the early detection [3]. Recently, it has been reported that lipid alterations such as the ratio of triglycerides to HDL cholesterol or pseudocholinesterase activity have an independent role in predicting the presence of bladder cancer [4].

Transurethral resection of bladder tumors (TURBT) is usually performed as the standard treatment [5]. However, catheter-related bladder discomfort (CRBD) often occurs after TURBT because of the requirement for large-diameter urinary catheters postoperatively [6]. The clinical characteristics of CRBD include burning sensation and discomfort in the suprapubic region, in addition to urinary urgency and frequency [7]. Moreover, CRBD symptoms may become aggravated, resulting in poor patient satisfaction, emergence agitation, and delayed recovery [8,9].

The main mechanism responsible for CRBD is the activation of muscarinic receptors triggered by an indwelling urinary catheter [8,10]. The activation of muscarinic receptors results in involuntary bladder contractions, thereby causing urinary symptoms. Another mechanism that causes CRBD is associated with elevated prostaglandin level [8,11]. Along with activated capsaicin-sensitive C fibers, increased prostaglandins in the bladder due to inflammation, obstruction, and mucosal damage lead to detrusor muscle contraction [12].

Muscarinic receptor antagonists (solifenacin, darifenacin, and trospium) and antimuscarinic agents (oxybutynin, ketamine, tolterodine, tramadol, and butylscopolamine) prohibit smooth muscle contraction, including the detrusor, and have been treatment options for CRBD as well as overactive bladder [10,13]. However, they have various adverse effects originating from their non-selective nature, such as dry mouth, sedation, constipation, nausea, and vomiting [13,14]. Consequently, the clinical use of these agents in the treatment of CRBD is not preferred. Instead, a recent study proved the effect of intraoperative infusion of magnesium in lowering CRBD severity after TURBT [6]. Magnesium plays a role in stabilizing smooth muscle contraction because it is involved in the active transport of calcium ions through the cell membrane [15]. Additionally, the preventive role of ketorolac in postoperative CRBD was demonstrated in another study [12].

Multimodal analgesia in an enhanced recovery program is based on the concept of combining various drugs acting differently to maximize analgesia through their synergistic effects [16]. The authors hypothesized that intraoperative infusion of magnesium and ketorolac, which act on the muscarinic and inflammatory mechanisms of CRBD, respectively, is superior to a single medication in preventing CRBD occurrence after TURBT. To the best of our knowledge, no study has evaluated the effect of co-administration of magnesium and ketorolac on CRBD. In this study, intravenous magnesium and ketorolac were infused in the experimental group during surgery, whereas only intravenous ketorolac was injected in the control group. We aimed to demonstrate that CRBD incidence after TURBT would be significantly lower in the experimental group than in the control group.

## 2. Materials and Methods

### 2.1. Ethical Considerations

This prospective, randomized, double-blind, placebo-controlled study was approved by the Institutional Review Board and Ethics Committee of Seoul St. Mary’s Hospital, The Catholic University of Korea, a tertiary academic teaching facility, on 26 October 2021 (approval number: KC21OISI0841). The study protocol was registered with the International Committee of Medical Journal Editors (Clinical Research Information Service, Republic of Korea, approval number: KCT0006875) before the study commenced. Written informed consent was obtained from all enrolled patients between December 2021 and June 2022.

### 2.2. Study Population and Randomization

Patients who underwent TURBT were screened for eligibility to participate in the study. The exclusion criteria for the study were as follows: age < 19 or ≥80 years; an American Society of Anesthesiologists Physical Status score ≥ 3; preoperative effective glomerular filtration rate < 60 mL/min/1.73 m^2^; hypermagnesemia (serum magnesium level > 3.0 mg/dL); hypocalcemia (serum calcium level < 8.6 mg/dL); atrioventricular block; neuromuscular disease; change in surgical plans; taking medication known to interact with magnesium; and/or known gastrointestinal ulcer or coagulopathy.

The eligible patients were randomly allocated into the control group (*n* = 50) or the experimental group (*n* = 50) using a sealed-envelope method before enrollment in the study. The allocations were concealed in opaque and identical envelopes and opened immediately before induction by an attending nurse who was responsible for preparing the study drugs: magnesium solution in the experimental group or normal saline in the control group.

### 2.3. Study Protocol

The day before surgery, the attending physician educated the study patients on CRBD symptoms (burning sensation with urinary urgency or discomfort in the suprapubic area) and how they were differentiated from surgical pain. Premedication was not administered to the patients. The attending anesthesiologists were blinded to group allocation, and anesthetic management was performed per the study protocol.

Standard multiple monitoring devices (electrocardiograph, pulse oximeter, non-invasive blood pressure measurement device, bispectral index assessment device, and esophageal thermometer) were placed in the operating room. Anesthesia was induced using propofol (1.5–2 mg/kg) and rocuronium (0.6 mg/kg). After confirmation of loss of consciousness and spontaneous respiration, I-gel^TM^ (Intersurgical Ltd., Berkshire, UK) insertion was performed.

The experimental group received intravenous infusion of magnesium (40 mL, containing 10% solution; Daehan Pharmaceutical Company, Seoul, Korea) after induction of anesthesia. As a loading dose, 50 mg/kg of magnesium was administered during the first 15 min. Magnesium was continuously infused at a rate of 15 mg/kg/h until the completion of surgery.

In the control group, 40 mL normal saline was continuously infused in the same manner.

In all patients, ketorolac (30 mg; Daewoo Pharmaceutical Company, Pusan, Korea) was injected after the induction of anesthesia. During surgery, desflurane (4–6%) was used to keep an adequate depth of anesthesia, with the bispectral index between 40 and 60. Remifentanil (0.02–0.10 µg/kg/min) was continuously infused to control surgical stimuli.

TURBT was performed by experienced surgeons. In the lithotomy position, a resectoscope was inserted into the bladder, and the entire bladder was examined. The confirmed mass lesions were electroresected, and the bleeding points were controlled with continuous irrigation. After sugammadex (4 mg/kg) was administered to reverse neuromuscular blockade and spontaneous eye-opening was confirmed, I-gel was removed under 100% O_2_ ventilation. At the end of the procedure, a 20-Fr, three-way urinary catheter was inserted, and the balloon was inflated with 10 mL of distilled water in every patient. To remove blood clots, continuous bladder irrigation was performed through the urinary catheter with normal saline. The urinary catheter was removed on postoperative day 1.

After the patient’s arrival at the post anesthesia care unit (PACU), fentanyl (50 µg; Daehan Pharmaceutical Company, Seoul, Korea) was administered if acute postoperative pain with a pain score > 30 mm on the visual analog scale (VAS) was noted. In the ward, patients were administered intravenous tramadol (50 mg; YUHAN, Seoul, Korea) as a rescue analgesic.

### 2.4. Study Outcomes

The primary outcome of the present study was CRBD incidence 0 h after TURBT, which was assessed upon the patient’s arrival at the PACU. CRBD incidence was also evaluated at 1 and 6 h after surgery. The attending physicians in charge of CRBD assessments were blinded to group assignment.

In addition, the patients were interviewed regarding CRBD severity and pain intensity. The degree of CRBD severity was assessed as follows: “mild” when reported by patients only upon questioning; “moderate” when reported by patients on their own without questioning and not accompanied by any behavioral response; and “severe” when reported by patients on their own with accompanying behavioral responses, such as flailing limbs, a strong vocal response, or an attempt to remove the catheter [17]. Postoperative pain intensity was evaluated using the VAS score (0 mm, no pain; 100 mm, worst possible pain) [18], and instances of rescue analgesic use were recorded. Side effects related to intraoperative magnesium infusion (headache, lethargy, flushing, and respiratory depression) and the occurrence of nausea and vomiting after surgery were also noted.

### 2.5. Clinical Variables

Demographic data including sex, age, body mass index, comorbidities (hypertension, diabetes mellitus, hepatitis B infection, and tuberculosis), American Society of Anesthesiologists physical status, and laboratory variables were collected. We also recorded intraoperative data, including the length of surgery, remifentanil dose, fluid input, and estimated blood loss. The blood loss was estimated by the operating surgeon who electro-resected the mass lesions and controlled the bleeding points under the guidance of resectoscope. In addition, information on multiplicity, size, shape and location of tumor was collected. Postoperatively, hospital stay duration and occurrence of clinical complications, which included significant hematuria requiring transfusion or transurethral intervention, were noted.

### 2.6. Sample Size and Statistical Analyses

Prior to the present study, we performed a preliminary study including 30 patients who underwent TURBT to assess CRBD incidence immediately after surgery. The incidence of early CRBD were 27% in the group that received magnesium and ketorolac during surgery (*n* = 15) and 60% in the group that received only ketorolac (*n* = 15). For a 5% risk of a type 1 error, a 10% risk of a type 2 error, and a dropout rate of 10%, 50 patients were required in each group.

The normality of continuous data was evaluated using the Shapiro–Wilk test. Descriptive statistics for categorical variables were reported as numbers (%), and those for continuous variables were reported as means (standard deviations) or medians (interquartile ranges). The χ^2^ or Fisher’s exact test was performed to compare categorical variables as appropriate. Student’s *t*-test or Mann–Whitney U test was used to compare continuous variables, as appropriate. All statistical tests were two-sided, and a *p*-value < 0.05 was considered to indicate significance. All analyses were performed using SPSS Statistics, version 24.0 (IBM Corp., Armonk, NY, USA).

## 3. Results

### 3.1. Comparisons of Preoperative Findings between the Control and Experimental Groups

A total of 144 patients were scheduled for elective TURBT during the study period (Figure 1). Among them, 5 refused to participate, 14 were older than 80 years, 13 had American Society of Anesthesiologists Physical Status scores ≥ 3, 7 had preoperative effective glomerular filtration rates < 60 mL/min/1.73 m^2^, and 5 with atrioventricular block did not meet the eligibility criteria for the study. Two patients in the experimental group underwent changes in the surgical plan during surgery and were excluded from the study. Thus, 50 patients in the control group and 48 in the experimental group completed the study.

Table 1 shows the preoperative findings in the two groups. No demographic or laboratory variables differed between the two groups.

### 3.2. Comparisons of Intraoperative Findings between the Control and Experimental Groups

The intraoperative findings of the two groups are shown in Table 2. Estimated blood loss was lower in the experimental group than in the control group (5 vs. 10 mL, *p* = 0.021), but it was not clinically meaningful.

### 3.3. Comparisons of the Incidence and Grade of Catheter-Related Bladder Discomfort after Surgery between the Control and Experimental Groups

As shown in Figure 2, the incidence of catheter-related bladder discomfort 0 h and 1 h after surgery was lower in the experimental than control group (31.3% vs. 52.0%, *p* = 0.037; 54.2% vs. 74.0%, *p* = 0.041, respectively). However, the incidence of catheter-related bladder discomfort 6 h after surgery and the grades of postoperative catheter-related bladder discomfort did not differ between the two groups (Table 3).

### 3.4. Comparisons of Postoperative Recovery Findings between the Control and Experimental Groups

Table 4 shows a comparison of the postoperative recovery findings between the two groups. There were no significant intergroup differences. No adverse effects associated with magnesium infusion (headache, lethargy, flushing, or respiratory depression) were observed in the study patients. All patients were discharged without clinical complications.

## 4. Discussion

The main finding of the study was that the co-administration of magnesium and ketorolac during surgery was superior to a single injection of ketorolac in preventing CRBD occurrence immediately after TURBT. The trend lasted until 1 h after surgery. However, the postoperative CRBD grades did not differ between the groups.

Effective analgesic methods after urological surgery have been reported in many studies [19,20]. However, clinicians are usually concerned about controlling parietal pain stemming from skin incisions or visceral pain related to intra-abdominal surgical manipulations. In addition, the mechanisms of CRBD differ from those of postoperative pain, resulting in conventional analgesics tolerance [10,12]. Due to these reasons, patients often remain undertreated for CRBD after surgery. CRBD occurrence, independent of postoperative pain, impairs the quality of recovery and triggers emergence agitation in patients in the PACU [10,21,22]. Therefore, we measured the total incidence of early CRBD after surgery as the primary outcome.

A multimodal approach for postoperative pain has been emphasized in many studies [23,24]. In particular, the enhanced recovery after surgery program emphasizes the use of non-opioid multimodal analgesics to control postoperative pain and to reduce opioid-related adverse effects [25,26]. Considering these perspectives, an effective management strategy for CRBD after surgery should be multimodal, utilizing non-opioid drugs. Thus, the present study was performed to determine the effectiveness of intraoperative administration of non-opioids in controlling CRBD after TURBT.

Many studies on atropine, dexmedetomidine, magnesium, paracetamol, ketorolac, and lidocaine have demonstrated their clinical efficacy in reducing CRBD after TURBT [6,14,27]. Among them, atropine, dexmedetomidine, and magnesium alleviated the occurrence and severity of CRBD after surgery due to their antimuscarinic properties. Paracetamol and ketorolac significantly reduced moderate-to-severe CRBD through their anti-inflammatory actions [11,12]. In contrast, lidocaine is considered to have both antimuscarinic and anti-inflammatory effects by interacting with sodium channels, muscarinic and N-methyl-d-aspartate receptors, and nociceptive transmission [28].

In this study, to determine the additional effects of magnesium in patients receiving ketorolac for CRBD control, magnesium and ketorolac were infused in the experimental group, while only ketorolac was injected in the control group. Ketorolac was selected based on its reported efficacy in preventing CRBD [12]. Magnesium was intravenously infused as an antimuscarinic agent in the experimental group. Other antimuscarinic agents, such as atropine and dexmedetomidine, were excluded from the study because they are related to changes in heart rate response or anesthetic depth in patients [29,30].

None of the patients in this study experienced magnesium-related side effects, such as lethargy, flushing, or respiratory depression. This is in concordance with the study by Park et al. [6] that compared magnesium infusion and normal saline for decreasing moderate-to-severe CRBD after TURBT. Additionally, our study indicated a lower incidence of moderate-to severe CRBD at 0 h postoperatively (10.4%), compared to Park et al.’s study [6] (22%). This superior effect is considered to be caused by the combination effect of magnesium and ketorolac.

Our study had several limitations. First, the individual effects of ketorolac in preventing CRBD were not assessed. Considering that the effect of ketorolac in decreasing CRBD has already been reported [12], the authors explored the additional effect of intraoperative magnesium infusion in patients receiving ketorolac. Second, generalizing the results of this study may be difficult because we evaluated the efficacy of the intraoperative administration of magnesium and ketorolac in older patients undergoing TURBT. Further studies in other populations with different surgical settings are required. Finally, there were no significant differences in the recovery profiles between the two groups although the incidence of early CRBD was significantly lower in the experimental group than in the control group. The present study may not be sufficiently powered to prove the enhanced patient recovery after surgery resulting from the effective control of early CRBD.

## 5. Conclusions

The present study demonstrated the efficacy of the intraoperative administration of magnesium and ketorolac in preventing early CRBD occurrence after TURBT. However, co-administration of the drugs did not reduce the grade of CRBD severity or the requirement for rescue analgesic. Future studies are required to investigate optimized combinations to control CRBD severity as well as CRBD incidence to promote patient recovery.

## Figures and Tables

**Figure 1 jcm-11-06359-f001:**
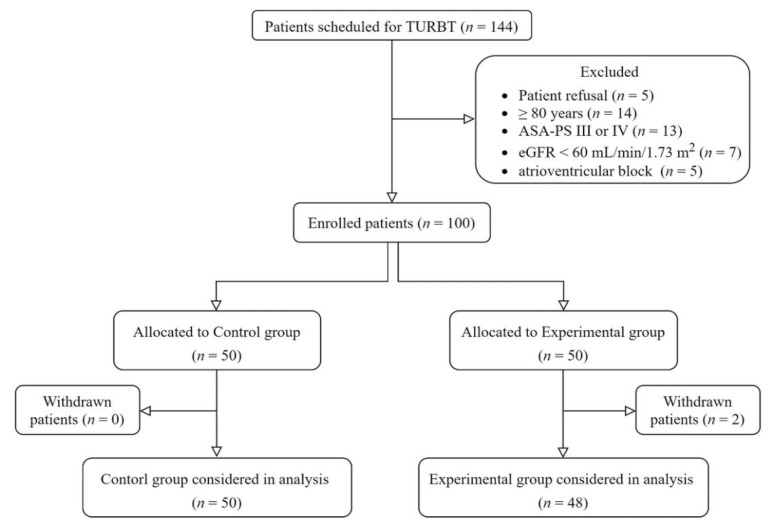
Study diagram. ASA-PS, American Society of Anesthesiologists Physical Status (classification system); eGFR, effective glomerular filtration rate; TURBT, transurethral resection of bladder tumor.

**Figure 2 jcm-11-06359-f002:**
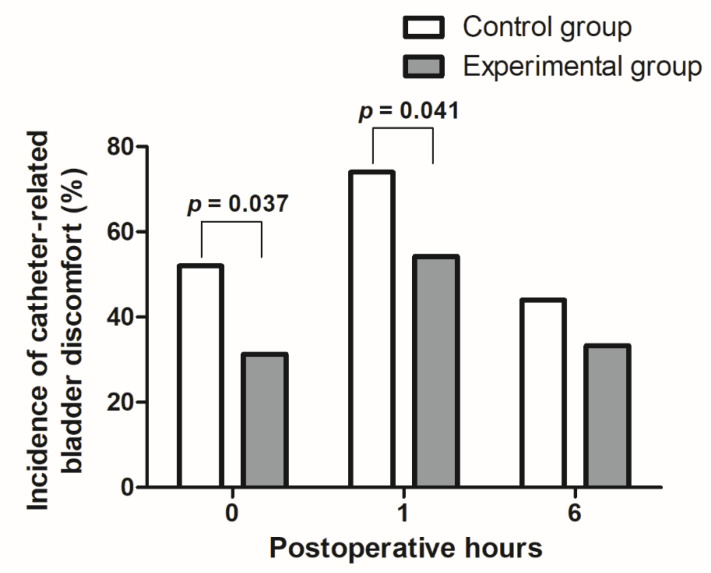
Comparison of catheter-related bladder discomfort incidences after surgery between the control and experimental groups.

**Table 1 jcm-11-06359-t001:** Comparisons of preoperative findings between the control and experimental groups.

	Control Group	Experimental Group	*p* Value
	(*n* = 50)	(*n* = 48)	
Sex (male)	40 (80.0%)	35 (72.9%)	0.408
Age (years)	66 (54–73)	68 (59–73)	0.325
Body mass index (kg/m^2^)	24.4 (22.3–25.9)	24.0 (22.3–26.6)	0.977
ASA classification			0.909
I/II	6/44	6/42	
Comorbidity			
Hypertension	19 (38.0%)	18 (37.5%)	0.959
Diabetes mellitus	8 (16.0%)	14 (29.2%)	0.118
Hepatitis B	4 (8.0%)	4 (8.3%)	>0.999
Tuberculosis	1 (2.0%)	4 (8.3%)	0.200
Laboratory variables			
White blood cell count (×10^9^/L)	6.7 (5.3–7.6)	6.3 (4.7–7.3)	0.320
Hemoglobin (g/dL)	14.3 (13.6–15.3)	14.0 (13.2–14.8)	0.091
Platelet count (×10^9^/L)	233 (192–284)	227 (190–289)	0.997
eGFR (mL/min/1.73 m^2^)	86.5 (75.6–101.3)	90.0 (78.9–102.4)	0.500
Calcium (mg/dL)	9.1 (8.8–9.4)	9.1 (8.8–9.5)	0.705
Magnesium (mg/dL)	2.2 (2.0–2.3)	2.1 (2.0–2.2)	0.440

Abbreviations: ASA, American Society of Anesthesiologists; eGFR, effective glomerular filtration rate. Note: Values are as median (interquartile) and number (proportions).

**Table 2 jcm-11-06359-t002:** Comparisons of intraoperative findings between the control and experimental groups.

	Control Group	Experimental Group	*p* Value
	(*n* = 50)	(*n* = 48)	
Total case length (min)	30 (25–38)	30 (25–39)	0.605
Remifentanil dose (mg)	0.1 (0.1–0.2)	0.1 (0.1–0.2)	0.624
IV fluids input (mL)	100 (50–100)	100 (50–100)	0.829
Estimated blood loss (mL)	10 (5–10)	5 (3–10)	0.021
Tumor multiplicity			0.430
single/multiple	20/30	23/25	
Tumor size			0.877
<1 cm	10 (20.0%)	11 (22.9%)	
1–3 cm	37 (74.0%)	35 (72.9%)	
>3 cm	3 (6.0%)	2 (4.2%)	
Tumor shape			0.795
Papillary	38 (76.0%)	38 (79.2%)	
Sessile	7 (14.0%)	7 (14.6%)	
Atypical	5 (10.0%)	3 (6.3%)	
Tumor location			
Dome	18 (36.0%)	15 (31.3%)	0.619
Anterior	10 (20.0%)	10 (20.8%)	0.918
Posterior	10 (20.0%)	13 (27.1%)	0.408
Right	11 (22.0%)	18 (37.5%)	0.093
Left	24 (48.0%)	15 (31.3%)	0.090
Trigon	6 (12.0%)	7 (14.6%)	0.706
Neck	8 (16.0%)	12 (25.0%)	0.269
Urethra	1 (2.0%)	1 (2.1%)	>0.999

NOTE: Values are expressed as median (interquartile) and number (proportions).

**Table 3 jcm-11-06359-t003:** Comparison of the grade of catheter-related bladder discomfort after surgery between the control and experimental groups.

	Control Group	Experimental Group	*p* Value
	(*n* = 50)	(*n* = 48)	
Postoperative hour 0			0.137
None	24 (48.0%)	33 (68.8%)	
Mild	15 (30.0%)	10 (20.8%)	
Moderate	9 (18.0%)	5 (10.4%)	
Severe	2 (4.0%)	0 (0.0%)	
Postoperative hour 1			0.180
None	13 (26.0%)	22 (45.8%)	
Mild	29 (58.0%)	21 (43.8%)	
Moderate	7 (14.0%)	5 (10.4%)	
Severe	1 (2.0%)	0 (0.0%)	
Postoperative hour 6			0.175
None	28 (56.0%)	32 (66.7%)	
Mild	19 (38.0%)	16 (33.3%)	
Moderate	3 (6.0%)	0 (0.0%)	
Severe	0 (0.0%)	0 (0.0%)	

NOTE: Values are expressed as number (proportions).

**Table 4 jcm-11-06359-t004:** Comparisons of postoperative recovery findings between the control and experimental groups.

	Control Group	Experimental Group	*p* Value
	(*n* = 50)	(*n* = 48)	
Postoperative pain score, Visual Analogue Scale			
Postoperative 0 h	2 (2–4)	2 (2–3)	0.361
Postoperative 1 h	2 (2–3)	2 (2–3)	0.452
Postoperative 6 h	2 (1–3)	2 (1–3)	0.669
Rescue analgesic use in PACU (%)	10 (20.0%)	4 (8.3%)	0.099
Rescue analgesic use in Ward (%)	6 (12.0%)	7 (14.6%)	0.706
Postoperative nausea/vomiting (%)	3 (6.0%)	3 (6.3%)	>0.999
Hospital stay, days	2 (2–2)	2 (2–2)	0.660

NOTE: Values are expressed as median (interquartile) and number (proportions).

## Data Availability

The data generated in this study can be shared after a reasonable request to the corresponding author.

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
