# Peer review of "Effects of Intraoperative Magnesium and Ketorolac on Catheter-Related Bladder Discomfort after Transurethral Bladder Tumor Resection: A Prospective Randomized Study"

_jcm, 2022, doi:10.3390/jcm11216359_

Round 1
Reviewer 1 Report
Dear Authors,
it is a pleasure to read an article of a research with such strong and sound materials and methods. Results are interesting, but might be improved their presentation, with a Table also reporting the different incidence of CRBD after TURB with the p-value expressed instead or in addition to Figure 2. Indeed, Table 3 and 4 in the current form do not display a statistically significant difference, thus might be misleading
Kind regards
Reviewer 2 Report
The authors presented a randomized study in which they examined the efficacy of the intraoperative administration of magnesium and ketorolac in preventing early CRBD occurrence after TURBT. There are some issues that should be addressed:
1. As reported catheter balloon volume could be a predictor of CRBD. What was the balloon volume in your study? Was it equal in all patients?
2. How was estimated blood loss calculated? Was there any difference in degree of hematuria in groups?
3. What was the size of tumors and bladder resected area? Was there any difference between groups?
4. How do you explain that catheter-related bladder discomfort 0 h and 1 h after surgery was lower in the experimental than control group when there was no difference in grade of CRBD, postoperative pain score and rescue analgesic usage between groups in this period? Are these findings clinically meaningful?
5. The conclusion should be rephrased according to the results of the study
Reviewer 3 Report
The authors should be congratulated for the interesting topic discussed and the great work done with this review.
The study design evaluated the efficacy of the intraoperative administration of magnesium and ketorolac only in older patients undergoing TURBT. This may reflect a selection bias. Whit this in mind, to generalize the results further studies in other populations with different surgical settings are required.
I believe that the study has sufficient merit, although major revisions are required.
1. Referred to study design: I believe that the CRBD and Postoperative pain intensity, evaluated using the VAS score, strongly depend on catheter size. I strongly recommend including this important missing information and if possible performing a subgroup analysis based on catheter size.
2. In this scenario, postoperative pain may depend on tumor size, resection area, and its width. Please include clinical tumor size if available.
3. Authors should discuss the mechanisms underlying the genesis and development of BC, even synthetically. In this regard, I recommend to reported BC pathological behavior. This article (PMCID: PMC9030452 - DOI: 10.3390/ijms23084173) offers an excellent examination of what is proposed. Among potential tumor biomarkers, Lipid alterations may play a kay role. In this regard, please include this article (PMCID: PMC8871224 - DOI: 10.3390/diagnostics12020431).
1. Please, add future perspective.
2. Check typos.
Round 2
Reviewer 2 Report
No further comments
Reviewer 3 Report
Authors answered all comments and suggestions.